# Proximal and Federated Random Reshuffling

## Abstract

Random Reshuffling (RR), also known as Stochastic Gradient Descent (SGD) without replacement, is a popular and theoretically grounded method for finite-sum minimization. We propose two new algorithms: Proximal and Federated Random Reshuffling (ProxRR and FedRR). The first algorithm, ProxRR, solves composite finite-sum minimization problems in which the objective is the sum of a (potentially non-smooth) convex regularizer and an average of $n$ smooth objectives. ProxRR evaluates the proximal operator once per epoch only. When the proximal operator is expensive to compute, this small difference makes ProxRR up to $n$ times faster than algorithms that evaluate the proximal operator in every iteration, such as proximal (stochastic) gradient descent. We give examples of practical optimization tasks where the proximal operator is difficult to compute and ProxRR has a clear advantage. One such task is federated or distributed optimization, where the evaluation of the proximal operator corresponds to communication across the network. We obtain our second algorithm, FedRR, as a special case of ProxRR applied to federated optimization, and prove it has a smaller communication footprint than either distributed gradient descent or Local SGD. Our theory covers both constant and decreasing stepsizes, and allows for importance resampling schemes that can improve conditioning, which may be of independent interest. Our theory covers both convex and nonconvex regimes. Finally, we corroborate our results with experiments on real data sets.

## 1 Introduction

Modern theory and practice of training supervised machine learning models is based on the paradigm of regularized empirical risk minimization (ERM) [Shalev-Shwartz and Ben-David, 2014]. While the ultimate goal of supervised learning is to train models that generalize well to unseen data, in practice only a finite data set is available during training. Settling for a model merely minimizing the average loss on this training set—the empirical risk—is insufficient, as this often leads to over-fitting and poor generalization performance in practice. Due to this reason, empirical risk is virtually always amended with a suitably chosen regularizer whose role is to encode prior knowledge about the learning task at hand, thus biasing the training algorithm towards better performing models.

The regularization framework is quite general and perhaps surprisingly it also allows us to consider methods for federated learning (FL)—a paradigm in which we aim at training model for a number of clients that do not want to reveal their data [Konečný et al., 2016, McMahan et al., 2017, Kairouz et al., 2019]. The training in FL usually happens on devices with only a small number of model updates being shared with a global host. To this end, Federated Averaging algorithm has emerged that performs Local SGD updates on the clients' devices and periodically aggregates their average. Its analysis usually requires special techniques and deliberately constructed sequences hindering the research in this direction. We shall see, however, that the convergence of our FedRR follows from merely applying our algorithm for regularized problems to a carefully chosen reformulation.

Formally, regularized ERM problems are optimization problems of the form

$$\min_{x \in \mathbb{R}^d} \left[ P(x) := \tfrac{1}{n} \sum_{i=1}^n f_i(x) + \psi(x) \right], \tag{1}$$

where $f_i \colon \mathbb{R}^d \to \mathbb{R}$ is the loss of model parameterized by vector $x \in \mathbb{R}^d$ on the $i$-th training data point, and $\psi \colon \mathbb{R}^d \to \mathbb{R} \cup \{+\infty\}$ is a regularizer. Let $[n] := \{1, 2, \ldots, n\}$. We shall make the following assumption throughout the paper without explicitly mentioning it:

**Assumption 1.** The functions $f_i$ are $L_i$-smooth, and the regularizer $\psi$ is proper, closed and convex. Let $L_{\max} := \max_{i \in [n]} L_i$.

In some results we will additionally assume that either the individual functions $f_i$, or their average $f := \tfrac{1}{n} \sum_i f_i$, or the regularizer $\psi$ are $\mu$-strongly convex. Whenever we need such additional assumptions, we will make this explicitly clear. While all these concepts are standard, we review them briefly in Section A.

**Proximal SGD.** When the number $n$ of training data points is huge, as is increasingly common in practice, the most efficient algorithms for solving (1) are stochastic first-order methods, such as stochastic gradient descent (SGD) [Bordes et al., 2009], in one or another of its many variants proposed in the last decade [Shang et al., 2018, Pham et al., 2020]. These method almost invariably rely on alternating stochastic gradient steps with the evaluation of the proximal operator

$$\mathrm{prox}_{\gamma\psi}(x) := \mathrm{argmin}_{z \in \mathbb{R}^d} \left\{ \gamma\psi(z) + \tfrac{1}{2}\|z - x\|^2 \right\}.$$

The simplest of these has the form

$$x_{k+1}^{\mathrm{SGD}} = \mathrm{prox}_{\gamma_k \psi}(x_k^{\mathrm{SGD}} - \gamma_k \nabla f_{i_k}(x_k^{\mathrm{SGD}})), \tag{2}$$

where $i_k$ is an index from $\{1, 2, \ldots, n\}$ chosen uniformly at random, and $\gamma_k > 0$ is a properly chosen learning rate. Our understanding of (2) is quite mature; see [Gorbunov et al., 2020] for a general treatment which considers methods of this form in conjunction with more advanced stochastic gradient estimators in place of $\nabla f_{i_k}$.

Applications such as training sparse linear models [Tibshirani, 1996], nonnegative matrix factorization [Lee and Seung, 1999], image deblurring [Rudin et al., 1992, Bredies et al., 2010], and training with group selection [Yuan and Lin, 2006] all rely on the use of hand-crafted regularizes. For most of them, the proximal operator can be evaluated efficiently, and SGD is near or at the top of the list of efficient training algorithms.

**Random reshuffling.** A particularly successful variant of SGD is based on the idea of random shuffling (permutation) of the training data followed by $n$ iterations of the form (2), with the index $i_k$ following the pre-selected permutation [Bottou, 2012]. This process is repeated several times, each time using a new freshly sampled random permutation of the data, and the resulting method is known under the name *Random Reshuffling (RR)*. When the same permutation is used throughout, the technique is known under the name *Shuffle-Once (SO)*.

One of the main advantages of this approach is rooted in its intrinsic ability to avoid cache misses when reading the data from memory, which enables a significantly faster implementation. Furthermore, RR is often observed to converge in fewer iterations than SGD in practice. This can intuitively be ascribed to the fact that while due to its sampling-with-replacement approach SGD can miss to learn from some data points in any given epoch, RR will learn from each data point in each epoch.

Understanding the random reshuffling trick, and why it works, has been a non-trivial open problem for a long time [Bottou, 2009, Recht and Ré, 2012, Gürbüzbalaban et al., 2019, Haochen and Sra, 2019]. Until recent development which lead to a significant simplification of the convergence analysis technique and proofs [Mishchenko et al., 2020], prior state of the art relied on long and elaborate proofs requiring sophisticated arguments and tools, such as analysis via the Wasserstein distance [Nagaraj et al., 2019], and relied on a significant number of strong assumptions about the objective [Shamir, 2016, Haochen and Sra, 2019]. In alternative recent development, Ahn et al. [2020] also develop new tools for analyzing the convergence of random reshuffling, in particular using decreasing stepsizes and for objectives satisfying the Polyak-Łojasiewicz condition, a generalization of strong convexity [Polyak, 1963, Lojasiewicz, 1963].

The difficulty of analyzing RR has been the main obstacle in the development of even some of the most seemingly benign extensions of the method. Indeed, while all these are well understood in

**Algorithm 1** Proximal Random Reshuffling (ProxRR) and Shuffle-Once (ProxSO)

**Require:** Stepsizes $\gamma_t > 0$, initial vector $x_0 \in \mathbb{R}^d$, number of epochs $T$
1: Sample a permutation $\pi = (\pi_{0u}, \pi_1, \ldots, \pi_{n-1})$ of $[n]$ (Do step 1 only for ProxSO)
2: **for** epochs $t = 0, 1, \ldots, T - 1$ **do**
3:     Sample a permutation $\pi = (\pi_0, \pi_1, \ldots, \pi_{n-1})$ of $[n]$ (Do step 3 only for ProxRR)
4:     $x_t^0 = x_t$
5:     **for** $i = 0, 1, \ldots, n - 1$ **do**
6:         $x_t^{i+1} = x_t^i - \gamma_t \nabla f_{\pi_i}(x_t^i)$
7:     $x_{t+1} = \text{prox}_{\gamma_t n \psi}(x_t^n)$

combination with its much simpler-to-analyze cousin SGD, *to the best of our knowledge, there exists no theoretical analysis of proximal, parallel, and importance sampling variants of RR with both constant and decreasing stepsizes, and in most cases it is not even clear how should such methods be constructed.* Empowered by and building on the recent advances of Mishchenko et al. [2020], in this paper we address all these challenges.

## 2 Contributions

In this section we outline the key contributions of our work, and also offer a few intuitive explanations motivating some of the development.

• **New algorithm: ProxRR.** Despite rich literature on Proximal SGD [Gorbunov et al., 2020], it is not obvious how one should extend RR to solve problem (1) when a regularizer $\psi$ is present. Indeed, the standard practice for SGD is to apply the proximal operator after each stochastic step [Duchi and Singer, 2009], i.e., in analogy with (2). On the other hand, RR is motivated by the fact that a data pass better approximates the full gradient step. If we applied the proximal operator after each step of RR, we would no longer approximate the full gradient after an epoch, as we illustrate next.

**Example 1.** Let $n = 2$, $\psi(x) = \frac{1}{2}\|x\|^2$, $f_1(x) = \langle c_1, x \rangle$, $f_2(x) = \langle c_2, x \rangle$ with some $c_1, c_2 \in \mathbb{R}^d$, $c_1 \neq c_2$. Let $x_0 \in \mathbb{R}^d$, $\gamma > 0$ and define $x_1 = x_0 - \gamma \nabla f_1(x_0)$, $x_2 = x_1 - \gamma \nabla f_2(x_1)$. Then, we have $\text{prox}_{2\gamma\psi}(x_2) = \text{prox}_{2\gamma\psi}(x_0 - 2\gamma \nabla f(x_0))$. However, if $\tilde{x}_1 = \text{prox}_{\gamma\psi}(x_0 - \gamma \nabla f_1(x_0))$ and $\tilde{x}_2 = \text{prox}_{\gamma\psi}(x_1 - \gamma \nabla f_2(\tilde{x}_1))$, then $\tilde{x}_2 \neq \text{prox}_{2\gamma\psi}(x_0 - 2\gamma \nabla f(x_0))$.

Motivated by this observation, we propose ProxRR (Algorithm 1), in which the proximal operator is applied at the end of each epoch of RR, i.e., after each pass through all randomly reshuffled data. A notable property of Algorithm 1 is that *only a single proximal operator evaluation is needed during each data pass.* This is in sharp contrast with the way Proximal SGD works, and offers significant advantages in regimes where the evaluation of the proximal mapping is expensive (e.g., comparable to the evaluation of $n$ gradients $\nabla f_1, \ldots, \nabla f_n$).

• **Convergence of ProxRR (for strongly convex functions or regularizer).** We establish several convergence results for ProxRR, of which we highlight two here. Both offer a linear convergence rate with a fixed stepsize to a neighborhood of the solution. In both we reply on Assumption 1. Firstly, in the case when in addition, each $f_i$ is $\mu$-strongly convex, we prove the rate (see Theorem 2)

$$\mathbb{E}\left[\|x_T - x_*\|^2\right] \leq (1 - \gamma\mu)^{nT} \|x_0 - x_*\|^2 + \frac{2\gamma^2 \sigma_{\text{rad}}^2}{\mu},$$

where $\gamma_t = \gamma \leq 1/L_{\max}$ is the stepsize, and $\sigma_{\text{rad}}^2$ is a *shuffling radius* constant (for precise definition, see (4)). In Theorem 1 we bound the shuffling radius in terms of $\|\nabla f(x_*)\|^2$, $n$, $L_{\max}$ and the more common quantity $\sigma_*^2 := \frac{1}{n}\sum_{i=1}^n \|\nabla f_i(x_*) - \nabla f(x_*)\|^2$. Secondly, if $\psi$ is $\mu$-strongly convex, and we choose the stepsize $\gamma_t = \gamma \leq 1/L_{\max}$, we prove the rate (see Theorem 3)

$$\mathbb{E}\left[\|x_T - x_*\|^2\right] \leq (1 + 2\gamma\mu n)^{-T} \|x_0 - x_*\|^2 + \frac{\gamma^2 \sigma_{\text{rad}}^2}{\mu}.$$

Both mentioned rates show exponential (linear in logarithmic scale) convergence to a neighborhood whose size is proportional to $\gamma^2 \sigma_{\text{rad}}^2$. Since we can choose $\gamma$ to be arbitrarily small or periodically

decrease it, this implies that the iterates converge to $x_*$ in the limit. Moreover, we show in Section 4 that when $\gamma = \mathcal{O}(\frac{1}{T})$ the error is $\mathcal{O}(\frac{1}{T^2})$, which is superior to the $\mathcal{O}(\frac{1}{T})$ error of SGD.

• **Results for SO.** All of our results apply to the Shuffle-Once algorithm as well. For simplicity, we center the discussion around RR, whose current theoretical guarantees in the nonconvex case are better than that of SO. Nevertheless, the other results are the same for both methods, and ProxRR is identical to ProxSO in terms of our theory too. A study of the empirical differences between RR and SO can be found in [Mishchenko et al., 2020].

• **Application to Federated Learning.** In Section 6 we describe an application of our results to federated learning [Konečný et al., 2016, McMahan et al., 2017, Kairouz et al., 2019]. In this way we obtain the FedRR method, which is similar to Local SGD, except the local solver is a single pass of RR over the local data. Empirically, FedRR can be vastly superior to Local SGD (see Figure 2). Remarkably, we also show that the rate of FedRR *beats the best known lower bound for Local SGD* due to [Woodworth et al., 2020] (we needed to adapt it from the original online to the finite-sum setting we consider in this paper) for large enough $n$. See Section F for more details.

• **Nonconvex analysis.** In the nonconvex regime, and under suitable assumptions, we establish (see Theorems 5 and 8) an $\mathcal{O}(\frac{1}{\gamma T})$ rate up to a neighborhood of size $\mathcal{O}(\gamma^2)$. For a certain stepsize it yields an $\mathcal{O}(\frac{1}{\varepsilon^3})$ convergence rate.

Besides the above results, we describe several extensions in the appendix, which we now outline.

• **Extension 1: Decreasing stepsizes.** The convergence of RR is not always exact and depends on the parameters of the objective. Similarly, if the shuffling radius $\sigma_{\mathrm{rad}}^2$ is positive, and we wish to find an $\varepsilon$-approximate solution, the optimal choice of a fixed stepsize for ProxRR will depend on $\varepsilon$. This deficiency can be fixed by using decreasing stepsizes in both vanilla RR [Ahn et al., 2020] and in SGD [Stich, 2019]. We adopt the same technique to our setting. However, we depart from [Ahn et al., 2020] by only adjusting the stepsize *once per epoch* rather than at every iteration, similarly to the concurrent work of Tran et al. [2020] on RR with momentum. For details, see Section I.

• **Extension 2: Importance resampling for Proximal RR.** While importance sampling is a well established technique for speeding up the convergence of SGD [Zhao and Zhang, 2015, Khaled and Richtárik, 2020], no importance sampling variant of RR has been proposed nor analyzed. This is not surprising since the key property of importance sampling in SGD—unbiasedness—does not hold for RR. Our approach to equip ProxRR with importance sampling is via a reformulation of problem (1) into a similar problem with a larger number of summands. In particular, for each $i \in [n]$ we include $n_i$ copies of the function $\frac{1}{n_i} f_i$, and then take average of all $N = \sum_i n_i$ functions constructed this way. The value of $n_i$ depends on the "importance" of $f_i$, described below. We then apply ProxRR to this reformulation. If $f_i$ is $L_i$-smooth for all $i \in [n]$ and we let $\bar{L} := \frac{1}{n} \sum_i L_i$, then we choose $n_i = \lceil L_i/\bar{L} \rceil$. It is easy to show that $N \leq 2n$, and hence our reformulation leads to at most a doubling of the number of functions forming the finite sum. However, the overall complexity of ProxRR applied to this reformulation will depend on $\bar{L}$ instead of $\max_i L_i$ (see Theorem 10), which can lead to a significant improvement. For details of the construction and our complexity results, see Section J.

## 3 Preliminaries

In our analysis, we build upon the notions of *limit points* and *shuffling variance* introduced by Mishchenko et al. [2020] for vanilla (i.e., non-proximal) RR. Given a stepsize $\gamma > 0$ (held constant during each epoch) and a permutation $\pi$ of $\{1, 2, \ldots, n\}$, the inner loop iterates of RR/SO converge to a neighborhood of intermediate limit points $x_*^1, x_*^2, \ldots, x_*^n$ defined by

$$x_*^i := x_* - \gamma \sum_{j=0}^{i-1} \nabla f_{\pi_j}(x_*), \quad i = 1, \ldots, n. \tag{3}$$

The intuition behind this definition is fairly simple: if we performed $i$ steps starting at $x_*$, we would end up close to $x_*^i$. To quantify the closeness, we define the *shuffling radius*.

**Definition 1** (Shuffling radius)**.** Given a stepsize $\gamma > 0$ and a random permutation $\pi$ of $\{1, 2, \ldots, n\}$ used in Algorithm 1, define $x_*^i = x_*^i(\gamma, \pi)$ as in (3). Then, the shuffling radius is defined by

$$\sigma_{\mathrm{rad}}^2(\gamma) := \max_{i=0,\ldots,n-1} \left[ \frac{1}{\gamma^2} \mathbb{E}_\pi \left[ D_{f_{\pi_i}}(x_*^i, x_*) \right] \right], \tag{4}$$

where the expectation is taken with respect to the randomness in the permutation $\pi$. If there are multiple stepsizes $\gamma_1, \gamma_2, \ldots$ used in Algorithm 1, we take the maximum of all of them as the shuffling radius, i.e., $\sigma_{\text{rad}}^2 := \max_{t \geq 1} \sigma_{\text{rad}}^2(\gamma_t)$.

The shuffling radius is related by a multiplicative factor in the stepsize to the shuffling variance introduced by Mishchenko et al. [2020]. When the stepsize is held fixed, the difference between the two notions is minimal. When the stepsize is decreasing, however, the shuffling radius is easier to work with, since it can be upper bounded by problem constants independent of the stepsizes.

Armed with a special lemma for sampling without replacement, we can upper bound the shuffling radius using the smoothness constant $L_{\max}$, size of the vector $\nabla f(x_*)$, and the variance $\sigma_*^2$ of the gradient vectors $\nabla f_1(x_*), \ldots, \nabla f_n(x_*)$.

**Theorem 1** (Bounding the shuffling radius). For any stepsize $\gamma > 0$ and any random permutation $\pi$ of $\{1, 2, \ldots, n\}$ we have $\sigma_{\text{rad}}^2 \leq \frac{L_{\max}}{2} n \left( n \|\nabla f(x_*)\|^2 + \frac{1}{2} \sigma_*^2 \right)$, where $x_*$ is a solution of Problem (1) and $\sigma_*^2$ is the population variance at the optimum

$$\sigma_*^2 := \frac{1}{n} \sum_{i=1}^{n} \|\nabla f_i(x_*) - \nabla f(x_*)\|^2. \tag{5}$$

All proofs are relegated to the supplementary material. In order to better understand the bound given by Theorem 1, note that if there is no proximal operator (i.e., $\psi = 0$) then $\nabla f(x_*) = 0$ and we get that $\sigma_{\text{rad}}^2 \leq \frac{L_{\max} n \sigma_*^2}{4}$. This recovers the existing upper bound on the shuffling variance of Mishchenko et al. [2020] for vanilla RR. On the other hand, if $\nabla f(x_*) \neq 0$ then we get an additive term of size proportional to the squared norm of $\nabla f(x_*)$.

# 4 Theory for strongly convex losses $f_1, \ldots, f_n$

Our first theorem establishes a convergence rate for Algorithm 1 applied with a constant stepsize to Problem (1) when each objective $f_i$ is strongly convex. This assumption is commonly satisfied in machine learning applications where each $f_i$ represents a regularized loss on some data points, as in $\ell_2$ regularized linear regression and $\ell_2$ regularized logistic regression.

**Theorem 2.** Let Assumption 1 be satisfied. Further, assume that each $f_i$ is $\mu$-strongly convex. If Algorithm 1 is run with constant stepsize $\gamma_t = \gamma \leq 1/L_{\max}$, then its iterates satisfy

$$\mathbb{E}\left[\|x_T - x_*\|^2\right] \leq (1 - \gamma\mu)^{nT} \|x_0 - x_*\|^2 + \frac{2\gamma^2 \sigma_{\text{rad}}^2}{\mu}.$$

We can convert the guarantee of Theorem 2 to a convergence rate by properly tuning the stepsize and using the upper bound of Theorem 1 on the shuffling radius. In particular, if we choose the stepsize as $\gamma = \min\left\{\frac{1}{L_{\max}}, \frac{\sqrt{\varepsilon\mu}}{\sqrt{2}\sigma_{\text{rad}}}\right\}$, and let $\kappa := L_{\max}/\mu$ and $r_0 := \|x_0 - x_*\|^2$, then we obtain $\mathbb{E}\left[\|x_T - x_*\|^2\right] = \mathcal{O}(\varepsilon)$ provided that the total number of iterations $K_{\text{RR}} = nT$ is at least

$$K_{\text{RR}} \geq \left[\left(\kappa + \frac{\sqrt{\kappa n}}{\sqrt{\varepsilon}\mu}(\sqrt{n}\|\nabla f(x_*)\| + \sigma_*)\right)\log\left(\frac{2r_0}{\varepsilon}\right)\right]. \tag{6}$$

**Comparison with vanilla RR.** If there is no proximal operator, then $\|\nabla f(x_*)\| = 0$ and we recover the earlier result of Mishchenko et al. [2020] on the convergence of RR without proximal, which is optimal in $\varepsilon$ up to logarithmic factors. On the other hand, when the proximal operator is nonzero, we get an extra term in the complexity proportional to $\|\nabla f(x_*)\|$: thus, even when all the functions are the same (i.e., $\sigma_* = 0$), we do not recover the linear convergence of Proximal Gradient Descent [Karimi et al., 2016, Beck, 2017]. This can be easily explained by the fact that Algorithm 1 performs $n$ gradient steps per one proximal step. Hence, even if $f_1 = \cdots = f_n$, Algorithm 1 does not reduce to Proximal Gradient Descent. We note that other algorithms for composite optimization which may not take a proximal step at every iteration (for example, using stochastic projection steps) also suffer from the same dependence [Patrascu and Irofti, 2021].

**Comparison with proximal SGD.** In order to compare (6) against the complexity of Proximal SGD (Algorithm 2), we recall that Proximal SGD achieves $\mathbb{E}\left[\|x_K - x_*\|^2\right] = \mathcal{O}(\varepsilon)$ if either $f$ or $\psi$ is $\mu$-strongly convex and

$$K_{\text{SGD}} \geq \left(\kappa + \frac{\sigma_*^2}{\varepsilon\mu^2}\right)\log\left(\frac{2r_0}{\varepsilon}\right). \tag{7}$$

---

**Algorithm 2** Proximal SGD

---

**Require:** Stepsizes $\gamma_k > 0$, initial vector $x_0 \in \mathbb{R}^d$, number of steps $K$
1: **for** steps $k = 0, 1, \ldots, K - 1$ **do**
2:     Sample $i_k$ uniformly at random from $[n]$
3:     $x_{k+1} = \text{prox}_{\gamma_k \psi}(x_k - \gamma_k \nabla f_{i_k}(x_k))$

---

This result is standard [Needell et al., 2016, Gower et al., 2019], with the exception that we do not know any proof in the literature for the case when $\psi$ is strongly convex. For completeness, we prove it in Appendix C, but since our proof is a minor modification of that in [Gower et al., 2019], we do not provide it here.

By comparing $K_{\text{SGD}}$ (given by (7)) and $K_{\text{RR}}$ (given by (6)), we see that ProxRR has milder dependence on $\varepsilon$ than Proximal SGD. In particular, ProxRR converges faster whenever the target accuracy $\varepsilon$ is small enough to satisfy $\varepsilon \leq \frac{1}{L_{\max} n \mu} \left( \frac{\sigma_*^4}{n \|\nabla f(x_*)\|^2 + \sigma_*^2} \right)$. Furthermore, ProxRR is much better when we consider *proximal iteration complexity* (# of proximal operator access), in which case the complexity of ProxRR (6) is reduced by a factor of $n$ (because we take one proximal step every $n$ iterations), while the proximal iteration complexity of Proximal SGD remains the same as (7). In this case, ProxRR is better whenever the accuracy $\varepsilon$ satisfies

$$\varepsilon \geq \frac{n}{L_{\max} \mu} \left[ n \|\nabla f(x_*)\|^2 + \sigma_*^2 \right] \qquad \text{or} \qquad \varepsilon \leq \frac{n}{L_{\max} \mu} \left[ \frac{\sigma_*^4}{n \|\nabla f(x_*)\|^2 + \sigma_*^2} \right].$$

We can see that if the target accuracy is large enough or small enough, and if the cost of proximal operators dominates the computation, ProxRR is much quicker to converge than Proximal SGD.

## 5   Theory for strongly convex regularizer $\psi$

In Theorem 2, we assume that each $f_i$ is $\mu$-strongly convex. This is motivated by the common practice of using $\ell_2$ regularization in machine learning. However, applying $\ell_2$ regularization in every step of Algorithm 1 can be expensive when the data are sparse and the iterates $x_t^i$ are dense, because it requires accessing each coordinate of $x_t^i$ which can be much more expensive than computing sparse gradients $\nabla f_i(x_t^i)$. Alternatively, we may instead choose to put the $\ell_2$ regularization inside $\psi$ and only ask that $\psi$ be strongly convex—this way, we can save a lot of time as we need to access each coordinate of the dense iterates $x_t^i$ only once per epoch rather than every iteration. Theorem 3 gives a convergence guarantee in this setting.

**Theorem 3.** Let Assumption 1 hold and $f_1, \ldots, f_n$ be convex. Further, assume that $\psi$ is $\mu$-strongly convex. If Algorithm 1 is run with constant stepsize $\gamma_t = \gamma \leq 1/L_{\max}$, where $L_{\max} = \max_i L_i$, then its iterates satisfy

$$\mathbb{E}\left[ \|x_T - x_*\|^2 \right] \leq (1 + 2\gamma\mu n)^{-T} \|x_0 - x_*\|^2 + \frac{\gamma^2 \sigma_{\text{rad}}^2}{\mu}.$$

Using Theorem 3 and choosing the stepsize as

$$\gamma = \min \left\{ \frac{1}{L_{\max}}, \frac{\sqrt{\varepsilon\mu}}{\sigma_{\text{rad}}} \right\}, \tag{8}$$

we get $\mathbb{E}\left[ \|x_T - x_*\|^2 \right] = \mathcal{O}(\varepsilon)$ provided that the total number of iterations satisfies

$$K \geq \left( \kappa + \frac{\sigma_{\text{rad}}/\mu}{\sqrt{\varepsilon\mu}} + n \right) \log \left( \frac{2r_0}{\varepsilon} \right). \tag{9}$$

This can be converted to a bound similar to (6) by using Theorem 1, in which case the only difference between the two cases is an extra $n \log \left( \frac{1}{\varepsilon} \right)$ term when only the regularizer $\psi$ is $\mu$-strongly convex. Since for small enough accuracies the $1/\sqrt{\varepsilon}$ term dominates, this difference is minimal.

## 6   FedRR: application of ProxRR to federated learning

Let us consider now the problem of minimizing the average of $N = \sum_{m=1}^{M} N_m$ functions that are stored on $M$ devices, which have $N_1, \ldots, N_M$ samples correspondingly,

$$\min_{x \in \mathbb{R}^d} F(x) + R(x), \qquad F(x) = \frac{1}{N} \sum_{m=1}^{M} F_m(x), \qquad F_m(x) = \sum_{j=1}^{N_m} f_{mj}(x). \tag{10}$$

---

**Algorithm 3** Federated Random Reshuffling (FedRR)

---

**Require:** Stepsize $\gamma > 0$, initial vector $x_0 = x_0^0 \in \mathbb{R}^d$, number of epochs $T$
1: **for** epochs $t = 0, 1, \ldots, T - 1$ **do**
2:     **for** $m = 1, \ldots, M$ locally in parallel **do**
3:         $x_{t,m}^0 = x_t$
4:         Sample permutation $\pi_{0,m}, \pi_{1,m}, \ldots, \pi_{N_m-1,m}$ of $\{1, 2, \ldots, N_m\}$
5:         **for** $i = 0, 1, \ldots, N_m - 1$ **do**
6:             $x_{t,m}^{i+1} = x_{t,m}^i - \gamma \nabla f_{\pi_{i,m}}(x_{t,m}^i)$
7:         $x_{t,m}^n = x_{t,m}^{N_m}$
8:     $x_{t+1} = \frac{1}{M} \sum_{m=1}^M x_{t,m}^n$

---

For example, $f_{mj}(x)$ can be the loss associated with a single sample $(X_{mj}, y_{mj})$, where pairs $(X_{mj}, y_{mj})$ follow a distribution $D_m$ that is specific to device $m$. An important instance of such formulation is federated learning, where $M$ devices train a shared model by communicating periodically with a server. We normalize the objective in (10) by $N$ as this is the total number of functions after we expand each $F_m$ into a sum. We denote the solution of (10) by $x_*$.

**Extending the space.** To rewrite the problem as an instance of (1), we are going to consider a bigger product space, which is sometimes used in distributed optimization [Bianchi et al., 2015]. Let us define $n := \max\{N_1, \ldots, N_m\}$ and introduce $\psi_C$, the *consensus* constraint, defined via

$$\psi_C(x_1, \ldots, x_M) := \begin{cases} 0, & x_1 = \cdots = x_M \\ +\infty, & \text{otherwise} \end{cases}.$$

By introducing dummy variables $x_1, \ldots, x_M$ and adding the constraint $x_1 = \cdots = x_M$, we arrive at the intermediate problem

$$\min_{x_1, \ldots, x_M \in \mathbb{R}^p} \frac{1}{N} \sum_{m=1}^M F_m(x_m) + (R + \psi_C)(x_1, \ldots, x_M),$$

where $R + \psi_C$ is defined, with a slight abuse of notation, as $(R + \psi_C)(x_1, \ldots, x_M) = R(x_1)$ if $x_1 = \cdots = x_M$, and $(R + \psi_C)(x_1, \ldots, x_M) = +\infty$ otherwise.

Since we have replaced $R$ with a more complicated regularizer $R + \psi_C$, we need to understand how to compute the proximal operator of the latter. We show (Lemma 7 in the supplementary) that the proximal operator of $(R + \psi_C)$ is merely the projection onto $\{(x_1, \ldots, x_M) \mid x_1 = \cdots = x_M\}$ followed by the proximal operator of $R$ with a smaller stepsize.

**Reformulation.** To have $n$ functions in every $F_m$, we write $F_m$ as a sum with extra $n - N_m$ zero functions, $f_{mj}(x) \equiv 0$ for any $j > N_m$, so that $F_m(x_m) = \sum_{j=1}^n f_{mj}(x_m) = \sum_{j=1}^{N_m} f_{mj}(x_m) + \sum_{j=N_m+1}^n 0$. We can now stick the vectors together into $\boldsymbol{x} = (x_1, \ldots, x_M) \in \mathbb{R}^{M \cdot d}$ and multiply the objective by $\frac{N}{n}$, which gives the following reformulation:

$$\min_{\boldsymbol{x} \in \mathbb{R}^{M \cdot d}} \frac{1}{n} \sum_{i=1}^n f_i(\boldsymbol{x}) + \psi(\boldsymbol{x}), \tag{11}$$

where $\psi(\boldsymbol{x}) := \frac{N}{n}(R + \psi_C)$ and

$$f_i(\boldsymbol{x}) = f_i(x_1, \ldots, x_M) := \sum_{m=1}^M f_{mi}(x_m).$$

In other words, function $f_i(\boldsymbol{x})$ includes $i$-th data sample from each device and contains at most one loss from every device, while $F_m(x)$ combines all data losses on device $m$. Note that the solution of (11) is $\boldsymbol{x}_* := (x_*^\top, \ldots, x_*^\top)^\top$ and the gradient of the extended function $f_i(\boldsymbol{x})$ is given by $\nabla f_i(\boldsymbol{x}) = (\nabla f_{1i}(x_1)^\top, \cdots, \nabla f_{Mi}(x_M)^\top)^\top$. Therefore, a stochastic gradient step that uses $\nabla f_i(\boldsymbol{x})$ corresponds to updating all local models with the gradient of $i$-th data sample, without any communication.

Algorithm 1 for this specific problem can be written in terms of $x_1, \ldots, x_M$, which results in Algorithm 3. Note that since $f_{mi}(x_i)$ depends only on $x_i$, computing its gradient does not require communication. Only once the local epochs are finished, the vectors are averaged as the result of projecting onto the set $\{(x_1, \ldots, x_M) \mid x_1 = \cdots = x_M\}$.

**Reformulation properties.** To analyze FedRR, the only thing that we need to do is understand the properties of the reformulation (11) and then apply Theorem 2 or Theorem 3. The following lemma gives us the smoothness and strong convexity properties of (11).

**Lemma 1.** Let function $f_{mi}$ be $L_i$-smooth and $\mu$-strongly convex for every $m$. Then, $f_i$ from reformulation (11) is $L_i$-smooth and $\mu$-strongly convex.

The previous lemma shows that the conditioning of the reformulation is $\kappa = \frac{L_{\max}}{\mu}$ just as we would expect. Moreover, it implies that the requirement on the stepsize remains exactly the same: $\gamma \leq 1/L_{\max}$. What remains unknown is the value of $\sigma_{\text{rad}}^2$, which plays a key role in the convergence bounds for ProxRR and ProxSO. To find an upper bound on $\sigma_{\text{rad}}^2$, let us define

$$\sigma_{m,*}^2 := \frac{1}{N_m} \sum_{j=1}^n \left\| \nabla f_{mj}(x_*) - \frac{1}{N_m} \nabla F_m(x_*) \right\|^2,$$

which is the variance of local gradients on device $m$. This quantity characterizes the convergence rate of local SGD [Yuan et al., 2020], so we should expect it to appear in our bounds too. The next lemma explains how to use it to upper bound $\sigma_{\text{rad}}^2$.

**Lemma 2.** The shuffling radius $\sigma_{\text{rad}}^2$ of the reformulation (11) is upper bounded by

$$\sigma_{\text{rad}}^2 \leq L_{\max} \cdot \sum_{m=1}^M \left( \|\nabla F_m(x_*)\|^2 + \frac{n}{4} \sigma_{m,*}^2 \right).$$

The lemma shows that the upper bound on $\sigma_{\text{rad}}^2$ depends on the sum of local variances $\sum_{m=1}^M \sigma_{m,*}^2$ as well as on the local gradient norms $\sum_{m=1}^M \|\nabla F_m(x_*)\|^2$. Both of these sums appear in the existing literature on convergence of Local GD/SGD [Khaled et al., 2019, Woodworth et al., 2020, Yuan et al., 2020]. We are now ready to present formal convergence results. For simplicity, we will consider heterogeneous and homogeneous cases separately and assume that $N_1 = \cdots = N_M = n$. To further illustrate generality of our results, we will present the heterogeneous assuming strong convexity $R$ and the homogeneous under strong convexity of functions $f_{mi}$.

**Heterogeneous data.** In the case when the data are heterogeneous, we provide the first local RR method. We can apply either Theorem 2 or Theorem 3, but for brevity, we give only the corollary obtained from Theorem 3.

**Theorem 4.** Assume that functions $f_{mi}$ are convex and $L_i$-smooth for each $m$ and $i$. If $R$ is $\mu$-strongly convex and $\gamma \leq 1/L_{\max}$, then we have for the iterates produced by Algorithm 3

$$\mathbb{E}\left[ \|x_T - x_*\|^2 \right] \leq (1 + 2\gamma\mu n)^{-T} \|x_0 - x_*\|^2 + \frac{\gamma^2 L_{\max}}{M\mu} \sum_{m=1}^M \left( \|\nabla F_m(x_*)\|^2 + \frac{N}{4M} \sigma_{m,*}^2 \right).$$

For nonconvex analysis, we consider $R \equiv 0$ and require the following standard assumption.

**Assumption 2** (Bounded variance and dissimilarity)**.** There exist constants $\sigma, \zeta > 0$ such that for any $x \in \mathbb{R}^d$ and

$$\frac{1}{n} \sum_{i=1}^n \left\| \nabla f_{mi} - \frac{1}{n} \nabla F_m(x) \right\|^2 \leq \sigma^2 \qquad \text{and} \qquad \frac{1}{M} \sum_{m=1}^M \left\| \frac{1}{n} \nabla F_m(x) - \nabla F(x) \right\|^2 \leq \zeta^2.$$

Note that above $\frac{1}{n} \nabla F_m(x) = \frac{1}{N_m} \nabla F_m(x)$ is the gradient of a local dataset and $\nabla F(x) = \frac{1}{N} \sum_{l=1}^M \nabla F_l(x)$ is the full gradient on all data.

**Theorem 5** (Nonconvex convergence)**.** Let Assumptions 1 and 2 be satisfied, and $R \equiv 0$ (no prox). Then, the communication complexity to achieve $\mathbb{E}\left[ \|\nabla F(x_T)\|^2 \right] \leq \varepsilon^2$ is

$$T = \mathcal{O}\left( \left( \frac{1}{\varepsilon^2} + \frac{\sigma}{\sqrt{n}\varepsilon^3} + \frac{\zeta}{\varepsilon^3} \right) (F(x_0) - F_*) \right).$$

Notice that by replicating the data locally on each device and thereby increasing the value of $n$ without changing the objective, we can improve the second term in the communication complexity. In particular, if the data are not too dissimilar ($\sigma \gg \zeta$) and $\varepsilon$ is small ($\frac{1}{\varepsilon^3} \gg \frac{1}{\varepsilon^2}$), the second term in the complexity dominates, and it helps to have more local steps. However, if the data are less similar, the nodes have to communicate more frequently to get more information about other objectives.

**Homogeneous data.** For simplicity, in the homogeneous (i.e., i.i.d.) data case we provide guarantees without the proximal operator. Since then we have $F_1(x) = \cdots = F_M(x)$, for any $m$ it holds $\nabla F_m(x_*) = 0$, and thus $\sigma_{m,*}^2 = \frac{1}{n} \sum_{j=1}^n \|\nabla f_{mj}(x_*)\|^2$. The full variance is then given by

$$\sum_{m=1}^M \sigma_{m,*}^2 = \frac{1}{n} \sum_{m=1}^M \sum_{i=1}^n \|\nabla f_{mi}(x_*)\|^2 = \frac{N}{n} \sigma_*^2 = M \sigma_*^2,$$

where $\sigma_*^2 := \frac{1}{N} \sum_{i=1}^n \sum_{m=1}^M \|\nabla f_{mi}(x_*)\|^2$ is the variance of the gradients over all data.

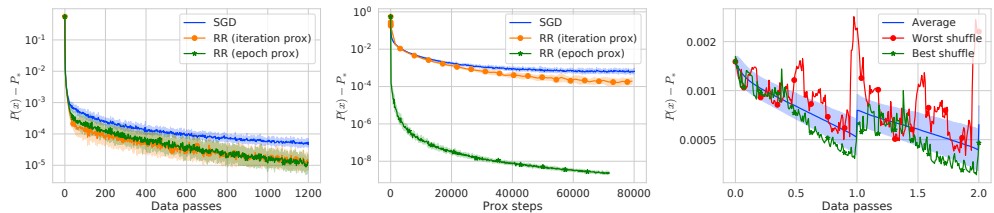

Figure 1: Experimental results for problem (12). The first two plots show with average and confidence intervals estimated on 20 random seeds and clearly demonstrate that one can save a lot of proximal operator computations with our method. The right plot shows the best/worst convergence of ProxSO over 20,000 sampled permutations.

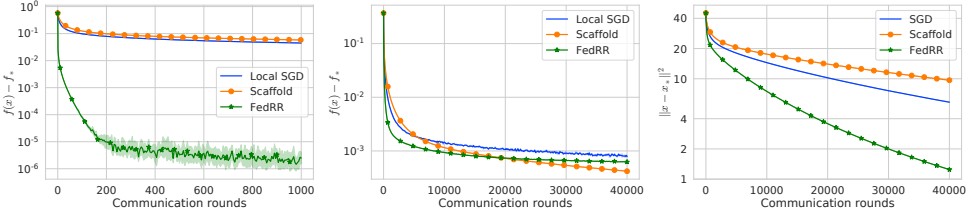

Figure 2: FedRR vs Local-SGD and Scaffold: i.i.d. data (left) and heterogeneous data (middle and right). We set $\lambda_1 = 0$ and estimate the averages and standard deviations by running 10 random seeds for each method.

**Theorem 6.** Let $R(x) \equiv 0$ (no prox) and the data be i.i.d., that is $\nabla F_m(x_*) = 0$ for any $m$, where $x_*$ is the solution of (10). Let $\sigma_*^2 := \frac{1}{N} \sum_{i=1}^{n} \sum_{m=1}^{M} \|\nabla f_{mi}(x_*)\|^2$. If each $f_{mj}$ is $L_{\max}$-smooth and $\mu$-strongly convex, then the iterates of Algorithm 3 satisfy

$$\mathbb{E}\left[\|x_T - x_*\|^2\right] \leq (1 - \gamma\mu)^{nT} \|x_0 - x_*\|^2 + \frac{\gamma^2 L_{\max} N \sigma_*^2}{M\mu}.$$

The most important part of this result is that the last term in Theorem 6 has a factor of $M$ in the denominator, meaning that the convergence bound improves with the number of devices involved.

## 7   Experiments[1]

**ProxRR vs SGD.** In Figure 1, we look at the logistic regression loss with the elastic net regularization,

$$\frac{1}{N} \sum_{i=1}^{N} f_i(x) + \lambda_1 \|x\|_1 + \frac{\lambda_2}{2} \|x\|^2, \tag{12}$$

where each $f_i : \mathbb{R}^d \to \mathbb{R}$ is defined as $f_i(x) := -\left(b_i \log\left(h(a_i^\top x)\right) + (1 - b_i) \log\left(1 - h(a_i^\top x)\right)\right)$, and where $(a_i, b_i) \in \mathbb{R}^d \times \{0, 1\}$, $i = 1, \ldots, N$ are the data samples, $h : t \to 1/(1 + e^{-t})$ is the sigmoid function, and $\lambda_1, \lambda_2 \geq 0$ are parameters. We set minibatch sizes to 32 for all methods and use theoretical stepsizes, without any tuning. We denote the heuristic version of RR that performs proximal operator step after each iteration as 'RR (iteration prox)'. From the experiments, we can see that all methods behave more or less the same way. However, the algorithm that we propose needs only a small fraction of proximal operator evaluations, which gives it a huge advantage whenever the operator takes more time to compute than stochastic gradients.

**FedRR vs Local SGD and Scaffold.** We also compare the performance of FedRR, Local SGD and Scaffold Karimireddy et al. [2020] on homogeneous (i.e., i.i.d.) and heterogeneous data. Since Local SGD and Scaffold require smaller stepsizes to converge, they are significantly slower in the i.i.d. regime, as can be seen in Figure 2. FedRR, however, does not need small initial stepsize and very quickly converges to a noisy neighborhood of the solution. We obtain heterogeneous regime by sorting data with respect to the labels and mixing the sorted dataset with the unsorted one. In this scenario, we also use the same small stepsize for every method to address the data heterogeneity. Clearly, Scaffold is the best in terms of functional values because it does variance reduction with respect to the data. Extending FedRR in the same way might be useful too, but this goes beyond the scope of our paper and we leave it for future work. We also note that in terms of distances from the optimum, FedRR still performs much better than Local SGD and Scaffold.

---

[1]Our code is provided in the supplementary. More experimental details are in the appendix.

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
