# OpenReview forum: "Proximal and Federated Random Reshuffling"
_NeurIPS.cc/2021/Conference — NeurIPS 2021 Submitted_

### Official Review · Reviewer_SKrH · 2021-07-13

**Rating:** 6
**Confidence:** 3

**Summary:**

This paper studies a proximal extension of random reshuffling algorithm, namely ProxRR for finite-sum composite optimization. ProxRR requires only one proximal steps per round, which reduces the complexities incurred by costly proximal operators. This paper also studies a federated version of ProxRR, namely FedRR, and proved theoretical advantages over Local SGD.

**Limitations And Societal Impact:**

See main review for limitations.

**Main Review:**

I would like to express my appreciation for your submission to NeurIPS 2021. The paper is overall well-written and I enjoyed reading it. My concerns are as follows.

1. The ProxRR presented in this paper applies only one proximal step per epoch. This reduces the number of calls to proximal operators, but at the same time incurs additional complexities (say, in eq (6)). It is hard to calibrate the significance of ProxRR bound (6) with [1] or Proximal SGD (7) since there are two factors mixed in --- reshuffling and reduced proximal calls. Have you considered doing ablation study to separate these two factors? For example, how does “ProxRR with proximal at every step” work? Will it eliminate the additional term in (6)?

2. This work compares  FedRR with other smooth federated optimization algorithm such as FedAvg and SCAFFOLD. How does FedRR compare with other federated composite algorithms such as [2] (theoretically or empirically)? For example, for experiments presented in Fig. 2, how does FedRR compare with FedDualAvg [2] if $\lambda_1 \neq 0$?

[1] Random Reshufﬂing: Simple Analysis with Vast Improvements, NeurIPS 2020.
[2] Federated Composite Optimization, ICML 2021.


**Time Spent Reviewing:**

2

---

> ### Author Response · Authors · 2021-08-10
> **All concerns handled**
>
> We thank the reviewer for the positive feedback and we are glad that the reviewer appreciated our work. Below we address both of your concerns and we hope that you would find them fully resolved:
> 1. It is a great question as there are indeed two components: reshuffling and reduced proximal calls. However, **we already did an equivalent of ablation studies** and **by comparing to RR** (which removes the effect of reduced prox calls) in Figure 1 **and by comparing to Local SGD** (which removes the effect of reshuffling) in Figure 2. We believe that the results are clear: Figure 1 shows that reduced prox calls (green line vs orange line) lead to slightly slower convergence at the beginning in terms of iterations and to a much better performance in terms of number of proximal operator calls. Figure 2, in turn, shows that the effect of reshuffling (blue line vs green line) depends on the problem and might be either large (left) or small (middle and right). We will explain in the paper how exactly each component of the proposed algorithm contributes to its performance.
> 2. Thank you for providing a highly relevant reference [2], which was published only recently. We cannot compare the theory directly because their work considers only the convex case and does not consider strongly convex or nonconvex case. We can only make a few observations. Firstly, Theorem 4.1 of [2] requires a much smaller stepsize $1/(K\cdot L)$, where $K$ is the number of local steps. Their Theorem 4.2 lifts this assumption and is more similar to our result, but it requires uniformly bounded gradients, while we measure convergence in terms of the gradient norm at the optimum. It is interesting to note that both our strongly convex result (6) and their bound (4.2) include (upper bound on) gradient norm. The key difference is of course that [2] uses an SGD-type algorithm, so their algorithms may not achieve the $O(1/T^2)$ rate of our algorithm in the strongly convex case, since even SGD converges as $O(1/T)$. We will add a numerical comparison with FedDualAvg in the setting $\lambda_1\neq 0$, as per your suggestion. We hope this completely addresses your concern.
>
> The reviewer's concerns appear to be minor and more like questions, both of them we answer. Otherwise, the reviewer gave the paper only positive feedback, so we are wondering why you gave the paper **the smallest** accept score?

---

> > ### Comment · Reviewer_SKrH · 2021-08-20
> > **Thanks for the response, and explaining question 1**
> >
> > Thanks for the response. In question 1 I am mostly looking for a "theoretical" ablation study: For example, how does “ProxRR with proximal at every step” work? Will it eliminate the additional term in (6)?
> >
> > I gave the "marginal above the acceptance threshold" score mostly because of the above concern: it is hard to calibrate the significance of ProxRR bound due to the mixed factors (especially considering this paper is largely theoretical). I am happy to re-evaluate the paper if the above concern can be resolved. Thanks!

---

> > > ### Author Response · Authors · 2021-08-25
> > > **New theoretical analysis for the case when prox is used every step**
> > >
> > > Thank you very much for responding to our rebuttal, we are happy that you read it.  We took time to write a detailed response to your question, so we hope that you would read it too and take into account when making a final decision about our paper.
> > > In addition to our empirical study, we provide the desired theoretical comparison. Before we proceed to the details, here is a short answer to your question: **if the proximal operator is evaluated every steps, there will be no gradient-norm term but the variance term will blow up and make it overall worse**.
> > >
> > > Now in more detail. With the proximal operator evaluated at each iteration, the update would be $x_t^{i+1} = prox_{\gamma \psi}(x_t^i - \gamma \nabla f_{\pi_i}(x_t^i))$, for which we were able to prove (the proof is given below) the following recursion in the strongly convex setting:
> > > $\Vert x_{t}^{n} - \hat x_\ast^n\Vert^2 \le (1-\gamma\mu)^n \Vert x_t^0 - x_\ast\Vert ^2 + 2\gamma^3 \hat \sigma_{rad}^2\sum_{j=0}^{n-1}(1-\gamma\mu)^j$
> > > where we define recursively $\hat x_\ast^{i+1} = prox_{\gamma \psi}(\hat x_\ast^i-\gamma \nabla f_{\pi_i}(x_\ast))$ and $\hat \sigma_{rad}$ is defined the same way as before (see equation (4) in our paper) but for this new sequence $\hat x_\ast^i$. **The important fact is that in general $\hat x_\ast^n\neq x_\ast$**. This almost breaks convergence as one has to use inequality $\Vert a+b\Vert ^2\le (1+\rho)\Vert a\Vert^2 + (1+\frac{1}{\rho})\Vert b\Vert^2$ to get
> > > $\Vert x_{t}^{n} -  x_\ast\Vert^2 \le (1+\rho)\left((1-\gamma\mu)^n \Vert x_t^0 - x_\ast\Vert ^2 + 2\gamma^3 \hat \sigma_{rad}^2\sum_{j=0}^{n-1}(1-\gamma\mu)^j \right) + (1+\frac{1}{\rho})\Vert \hat x_\ast^n - x_\ast\Vert^2$
> > > $\le (1+\rho)\left((1-\gamma\mu)^n \Vert x_t^0 - x_\ast\Vert ^2 + 2\gamma^3 \hat \sigma_{rad}^2\sum_{j=0}^{n-1}(1-\gamma\mu)^j \right) + (1+\frac{1}{\rho})\gamma^2\sigma_\ast^2 n.$
> > > This upper bound is immediately problematic because the last term is $O(\gamma^2)$ and not $O(\gamma^3)$, so we cannot hope for $O(1/k^2)$ convergence, only $O(1/k)$ rate is possible (following the standard proof of SGD). However, things might be even worse since we can show that the coefficient $1+\frac{1}{\rho}$ might be huge (see the end of this post if interested). To the best of our knowledge, the proofs given below are completely new.
> > >
> > > *Proof of the recursion*.
> > > By nonexpansiveness of the proximal operator (see Lemma 5 in our paper), we have
> > > $\Vert x_t^{i+1} - \hat x_\ast^{i+1}\Vert ^2 = \Vert prox_{\gamma \psi}(x_t^i - \gamma \nabla f_{\pi_i}(x_t^i)) - prox_{\gamma \psi}(\hat x_\ast^i - \gamma \nabla f_{\pi_i}(x_\ast))\Vert ^2$
> > > $\le \Vert x_t^i - \gamma \nabla f_{\pi_i}(x_t^i) - (\hat x_\ast^i - \gamma \nabla f_{\pi_i}(x_\ast))\Vert ^2$
> > > $= \Vert x_t^{i} - \hat x_\ast^{i}\Vert ^2 - 2\gamma\langle \nabla f_{\pi_i}(x_t^i) - \nabla f_{\pi_i}(x_\ast), \hat x_t^i - x_\ast \rangle + \gamma^2 \Vert \nabla f_{\pi_i}(x_t^i) - \nabla f_{\pi_i}(x_\ast)\Vert^2$
> > > $= \Vert x_t^{i} - \hat x_\ast^{i}\Vert ^2 - 2\gamma[D_{f_{\pi_i}}(x_t^{i}, x_\ast) + D_{f_{\pi_i}}(\hat x_\ast^{i}, x_t^i) - D_{f_{\pi_i}}(\hat x_\ast^i, x_\ast)] + \gamma^2 \Vert \nabla f_{\pi_i}(x_t^i) - \nabla f_{\pi_i}(x_\ast)\Vert^2\quad $  (three-point identity)
> > > $= \Vert x_t^{i} - \hat x_\ast^{i}\Vert ^2 - 2\gamma[D_{f_{\pi_i}}(x_t^{i}, x_\ast) + D_{f_{\pi_i}}(\hat x_\ast^{i}, x_t^i) - D_{f_{\pi_i}}(\hat x_\ast^i, x_\ast)] + 2\gamma^2 L D_{f_{\pi_i}}(x_t^{i}, x_\ast)\quad $ ($L$-smoothness of $f_{\pi_i}$)
> > > $\le (1-\gamma\mu)\Vert x_t^{i} - \hat x_\ast^{i}\Vert ^2 + 2\gamma D_{f_{\pi_i}}(\hat x_\ast^i, x_\ast)\quad $    (by $\gamma\le\frac{1}{L}$ and $\mu$-strong convexity of $f_{\pi_i}$)
> > > $\le (1-\gamma\mu)\Vert x_t^{i} - \hat x_\ast^{i}\Vert ^2 + \gamma L\Vert \hat x_\ast^i - x_\ast\Vert^2\quad$ ($L$-smoothness of $f_{\pi_i}$)
> > >
> > > *Proof of the upper bound on $\Vert \hat x_\ast^n  - x_\ast\Vert$*
> > > We obtain by nonexpansiveness of the proximal operator and triangle inequality
> > > $\Vert \hat x_\ast^{i+1} - x_\ast\Vert \le \Vert \hat x_\ast^{i} - x_\ast - \gamma( \nabla f_{\pi_i}(x_\ast) - \nabla f(x_\ast)) \Vert \le \Vert \hat x_\ast^{i} - x_\ast\Vert + \gamma\Vert \nabla f_{\pi_i}(x_\ast) - \nabla f(x_\ast) \Vert$
> > > So by induction and AM-QM inequality it holds
> > > $\Vert \hat x_\ast^{n} - x_\ast\Vert \le \gamma\sum_{j=0}^{n-1} \Vert \nabla f_{\pi_j}(x_\ast) - \nabla f(x_\ast) \Vert \le \gamma n\sqrt{\frac{1}{n}\sum_{j=0}^{n-1} \Vert \nabla f_{\pi_j}(x_\ast) - \nabla f(x_\ast) \Vert^2} = \gamma \sqrt{n} \sigma_\ast$.
> > >
> > > *Proof that the extra term is problematic*
> > > To keep the contraction, we have to use $\rho$ such that $(1+\rho)(1-\gamma \mu)^n < (1-\gamma\mu/2)^n$, which is tightly satisfied if we choose $\rho=(1+\gamma\mu/2)^n - 1$. For small $\gamma\mu$, this choice leads to $\rho\sim n\gamma\mu$ and $(1+\frac{1}{\rho})\gamma^2\sigma_\ast^2 n \sim \gamma \frac{\sigma_\ast^2}{\mu}$. After we recurse all the way to $\Vert x_0  - x_\ast\Vert^2$, this term would get divided by $\gamma\mu n$ one more time, giving an error term $O(\frac{\sigma_\ast^2}{\mu^2 n})$. Since it doesn't depend on $\gamma$, we can't make it smaller than some given $\varepsilon$ by decreasing $\gamma$.
> > >
> > > We hope that all steps are presented clearly and we will try to quickly respond to any questions about them.

---

### Official Review · Reviewer_Ak2z · 2021-07-14

**Rating:** 5
**Confidence:** 5

**Summary:**

This paper first proposes a proximal variant of the gradient method with shuffling strategy for a class of composite finite-sum minimization problems, and then specifies it for an optimization problem in federated learning (FL). In fact, the proposed algorithm performs n gradient descent updates per epoch and then applies the proximal operator at the end of each epoch. Both single shuffling and randomized reshuffling strategies can be used. The authors analyze the convergence rates of this algorithm under the strong convexity of the losses and the strong convexity of the regularizer. Next, they apply the proposed algorithm to FL, with and without regularizer. This variant is similar to FedAvg, but it replaces SGD by a shuffling gradient scheme for local updates. They establish the convergence of this variant for both the strongly convex case (where the regularizer is strongly convex) and the non-convex case in both heterogeneity and homogeneity settings, respectively. Numerical experiments are conducted to illustrate the algorithm and compare it with two existing methods: local SGD and SCAFFOLD.

**Ethics Review Area:**

["I don’t know"]

**Limitations And Societal Impact:**

Not discussed.

--- I would suggest the authors to compare also single shuffling (like incremental) and randomized reshuffling in Supp. Doc.



**Main Review:**

Originality: The first part of this paper extends the results in [Mishchenko et al, 2020] to the composite setting. The convergence results are very similar to those in this work. The use of proximal operator at the end of each epoch makes sense since the convergence guarantee will be in epoch, and it is independent of the shuffling strategy. This is perhaps one of the main contributions. Algorithm 3 is very similar to FedAvg by replacing a classical SGD update with a shuffling gradient scheme. In this algorithm, I do not see where the proximal operator of R comes in. I believe that it should be placed after line 8. Overall, the originality of this paper is moderate. Except for adding a new proximal step, the results are very similar to [Mishchenko et al, 2020]. In fact, [Mishchenko et al, 2020] essentially extends and improves [Nguyen et al 2020] to randomized reshuffling variants.

Quality/clarity: The paper is well written and well-motivated in general. The extension to composite form is nontrivial as they have discussed. However, I have a feeling that the analysis for the strongly convex case is not much different from [Mishchenko et al, 2020]. Therefore, the quality of the paper is moderate.

Significance: Overall, the paper has considerable contributions in terms of new algorithms, especially Algorithm 1, and the theoretical results associated with both Algorithms 1 and 3. But as mentioned above, due to the similarity of this paper and [Mishchenko et al, 2020], its significance is weak.

Below are some additional comments:

-- The authors only consider the strongly convex case for Algorithm 1, but not the convex and nonconvex cases. Why are the two latter cases not considered? Is the latter case a special case of Theorem 5?

-- In Theorem 2, how can one estimate or upper bound sigma_rad to form a learning rate? In Theorem 1, its upper bound depends on x*? I believe that this is for theoretical guarantee, but it seems to be very small since it depends on both sigma_rad and epsilon (a desired accuracy). Could you please comment on this aspect?

-- Is there any difference in terms of convergence if either the regularizer is strongly convex or at least one loss is strongly convex?

-- Is Theorem 4 a special case of Theorem 3?

-- In terms of analysis, what is the essential difference between [Mishchenko et al, 2020] and this paper when a regularizer is added?
Without a regularizer, does this analysis reduce to [Mishchenko et al, 2020] or other previous works?

-- The proof of Lemma 7 is unclear to me, making me unable to verify Algorithm 3. Could you please add more details to clarify its proof?

-- In Supp. Doc. the authors present the convergence for diminishing learning rates, when t > t_0 ~=T/2. Can you use diminishing learning rate from t=1? If so, then how does the convergence rate in Theorem 9 change?


**Time Spent Reviewing:**

8

---

> ### Author Response · Authors · 2021-08-10
> **The reviewer missed a key contribution -- the nonconvex analysis**
>
> We thank the reviewer for their thorough review and the effort taken to understand our work. One of the key points of our work was missed by the reviewer, in particular the presence of nonconvex analysis for Algorithm 1, so we hope the reviewer will be as committed to reading our feedback as they were to reading the paper.
> 1. “*I do not see where the proximal operator of R comes in*”
> In the presentation of FedRR, we omitted the proximal operator as we didn’t have enough space to discuss the corresponding results. The reviewer is right that it should be after step 8 if it was to be included. If the paper gets accepted, we will use the extra page in the camera-ready version to return R back to FedRR.
> 2. “*the analysis for the strongly convex case is not much different from [Mishchenko et al, 2020]. Therefore, the quality of the paper is moderate*”
> We believe that this conclusion is based on the wrong assumption that we don’t have nonconvex analysis, which is illustrated by another comment of the same reviewer “but not the convex and nonconvex cases”. We do analyze Algorithm 1 for nonconvex case in the appendix and **our nonconvex analysis is significantly different from the related work** on proximal algorithms and more challenging than the strongly convex one.
> 3. “*due to the similarity of this paper and [Mishchenko et al, 2020], its significance is weak*”
> Again, we want to emphasize that **our nonconvex analysis is a nontrivial extension of prior work on RR** and we are not aware of its analogues in the literature.
> 4. “*The authors only consider the strongly convex case for Algorithm 1*”
> This is plain wrong. We mention the nonconvex analysis in the “Contributions” section, and **we find that most of the reviewer’s criticism comes from this misunderstanding**.
> 5. “*how can one estimate or upper bound sigma_rad to form a learning rate*?”
> This constant does not need to be estimated if one uses decreasing stepsizes, which we develop in the appendix.
> 6. *“Is there any difference in terms of convergence if either the regularizer is strongly convex or at least one loss is strongly convex?”*
> The only minor difference is in the constants.
> 7. *“Is Theorem 4 a special case of Theorem 3?”*
> Yes, as explained in the last sentence before Theorem 4, it is obtained by applying the results in Theorem 3 to the reformulation from Section 6, whose properties are established in the appendix.
> 8. *“In terms of analysis, what is the essential difference between [Mishchenko et al, 2020] and this paper when a regularizer is added? Without a regularizer, does this analysis reduce to [Mishchenko et al, 2020] or other previous works?”*
> This is a great question. Our strongly convex analysis adds only a few steps to the one of [Mishchenko et al., 2020]. However, **our nonconvex analysis requires multiple extra steps** that do not include proximal operator, in particular all steps related to vector w_t are new. Without a regularizer, **the nonconvex analysis does not reduce to the one of [Mishchenko et al., 2020]**. Similarly, if one removes randomness by setting $f_1=\dotsb=f_n$, **the analysis does not reduce to the one of Proximal Gradient** either because we have to take care of multiple gradient steps per one proximal operator evaluation.
> 9. *“The proof of Lemma 7 is unclear to me, making me unable to verify Algorithm 3. Could you please add more details to clarify its proof?”*
> We will include more details to clarify the proof. Meanwhile, we did an extensive search of the literature on proximal operators to try and find a reference that would convince the reviewer that Lemma 7 is correct. You can consult Techinque 1 (the part about $prox_{\gamma f}$) in the paper [Condat et al., “​​Proximal Splitting Algorithms: A Tour of Recent Advances, with New Twists”], which gives Lemma 7 if one sets $\omega_1 = \dotsb = \omega_M =\frac{1}{M}$.
> 10. *“In Supp. Doc. the authors present the convergence for diminishing learning rates, when t > t_0 ~=T/2. Can you use diminishing learning rate from t=1? If so, then how does the convergence rate in Theorem 9 change?”*
> This is a great question! Indeed, we can use diminishing learnin rate from $t=1$, in which case the leading $O(1/T^2)$ term will not change. At the same time, the exponential term in Theorem 9 would change to a $O(1/T^3)$ term. In practice, we also noticed that running RR for a number of iterations without decreasing the stepsize sometimes helps in the initial phase.

---

> > ### Comment · Reviewer_Ak2z · 2021-08-24
> > **Thank you for your rebuttal.**
> >
> > Thank you very much for the rebuttal. I have read it carefully.
> >
> > Please note that in the summary, I already stated that you do have non-convex analysis. However, I did not raise any comments there due to time limitation.
> >
> > My concern in Item 4 is for Algorithm 1. In fact, I did not see where the analysis for the nonconvex case of Algorithm 1 is. In Sup. Doc. G, it seems that the analysis is for the federated learning reformulation (11), without the regularizer R, and therefore it should be Algorithm 3. Could you please tell me what I was missing?
> >
> > In addition, what is the fundamental difference between the new analysis of the nonconvex case compared to [Mishchenko et al., 2020]?
> > I find that one of the key bounds is in lines 790-791, which is very similar to Lemma 5 in [Mishchenko et al., 2020].
> > If you can provide your feedback in detail, then it would be great.

---

> > > ### Author Response · Authors · 2021-08-25
> > > **More about nonconvex analysis**
> > >
> > > We thank the reviewer for reading and responding to our rebuttal, we really appreciate it. We are happy to provide further clarifications regarding the nonconvex analysis:
> > > 1. Thank you for the clarification that your concern is about Algorithm 1. We do provide nonconvex guarantees for Algorithm 1 in Theorem 8 on page 27, we understand that it's easy to miss since it is in the appendix. You're right that Sup. Doc. G analyzes **reformulation (11) and Algorithm 3 in Section G.3**. However, please notice that **Sections G.1 and G.2 are for Algorithm 1**. For instance, Definition 2 on page 24 describes the proximal-gradient mapping that we used to state the guarantees for Algorithm 1.
> > > 2. The reviewer is correct that in lines 790-791 we have a similar step to the analysis of [Mishchenko et al., 2020], but **this is only the first part of the proof**. **Immediately after that, the analysis goes in a different direction** than that of [Mishchenko et al., 2020] and has multiple arguments different from it. The main difference comes in that we define a new sequence $w_t$ after line 796 and then produce a sequence of inequality to firstly upper bound $f(w_t)$ with $f(x_t)$, and then to upper bound $f(x_{t+1})$ with $f(w_t)$. See, for instance, the two inequalities that start right after line 806. The result of Theorem 8 is also slightly different from the prior work on random reshuffling as it depends on the newly introduced quantity $\zeta$ (notice the second term in the statement of Theorem 8), which corresponds to the data dissimilarity in the case of reformulation (11). While the first part of the proof takes similar steps as prior work, the second part was considerably more difficult and required us to reverse-engineer the bounded dissimilarity assumption from the literature on federated learning to understand what approach to use to analyze Algorithm 1 (and we particularly did not want to use bounded-gradients assumption, which could make the analysis much easier but less meaningful).

---

### Official Review · Reviewer_Gcuo · 2021-07-18

**Rating:** 5
**Confidence:** 3

**Summary:**

In this paper, the authors propose two new algorithms for finite sum minimization: Proximal Random Reshuffling (ProxRR) and Federated Random Reshuffling (FedRR). ProxRR is used to solve composite functions, while FedRR is a special case of ProxRR as used in the context of federated optimization. They provide convergence rates under various convexity assumptions, and they also show convergence to small gradient norm (in expectation) for smooth non-convex objectives.

**Limitations And Societal Impact:**

Yes.

**Main Review:**

In extending random reshuffling methods to composite optimization, the authors observe the importance of only applying the proximal operator once every epoch of random reshuffling, rather than at the end of each step. Thus, their algorithm only evaluates the proximal operator a single time during each pass of the data. They go on to show that their results apply to the setting of Shuffle-Once, which was also considered (alongside random reshuffling) in Mishchenko et al. (2020). The authors also provide promising experiments in the federated setting.

That said, the results of this paper, particularly in handling the random reshuffling difficulties, are somewhat of an incremental extension when compared to Mishchenko et al. (2020). It would be recommended that the authors better elaborate in what ways their results go beyond simply combining Mishchenko et al. (2020) with the standard prox analysis.


==== Minor comments ====

The authors should refer to, and compare with, the method FedAc [1].

Line 263: The equation is overlapping the text.

The authors note there is no tuning for the ProxRR vs SGD experiments (though tuning here would be *strongly* recommended), but what about for the federated optimization experiments? Please elaborate.


[1] Honglin Yuan, and Tengyu Ma. "Federated Accelerated Stochastic Gradient Descent." In NeurIPS. 2020.

**Time Spent Reviewing:**

4

---

> ### Author Response · Authors · 2021-08-10
> **Our nonconvex analysis is significantly different from any prior work**
>
> We thank the reviewer for their constructive feedback. We did not find any major criticism in the review, so we are confused by the low rating, but we still address the given comments below.
> 1. *“the authors better elaborate in what ways their results go beyond simply combining Mishchenko et al. (2020) with the standard prox analysis”*
> First of all, even if we remove RR from consideration by setting $f_1=f_2=\dotsb=f_n$, we still recover a **new** deterministic algorithm! We are not aware of any analysis for proximal gradient descent with the proximal operator computed every $n$ steps. **The main difficulty**, however, is in our **nonconvex analysis**. Proving convergence of Proximal SGD in the nonconvex regime took the community a considerable effort. It was solved in the paper (Davis and Drusvyatskiy, “Stochastic model-based minimization of weakly convex functions”) in 2018 and the work has already 177 citations. However, their approach significantly relies on **unbiased updates**, which **cannot be used for RR**. Therefore, **we introduced Assumption 3**, which is, to the best of our knowledge, new, and it is very natural since it recovers as a special case the **widely adopted bounded dissimilarity assumption** that can be found in the analysis of Local SGD and other local methods. We believe that **these aspects significantly distinguish our work** from the rest of papers with prox analysis.
> 2. Adding a comparison to FedAc is a great idea! In the strongly convex case, FedAc improves upon FedAvg only in one of the sublinear terms and has overall $O(1/T)$ rate (for simplicity, let us ignore all constants). Our analysis, however, gives $O(1/T^2$), so **FedRR is asymptotically faster than FedAc**. In some target-accuracy regimes, FedAc might still be faster, which is similar to how accelerated SGD can be faster than RR. We believe it is **not an issue** because one could try to **accelerate our algorithm** too. Notice that we also analyze nonconvex case, which is not covered by FedAc.
> 3. *“tuning here would be strongly recommended <...> what about for the federated optimization experiments?”*
> The purpose of our experiments is to elaborate on the theory and not to convince the reader that the proposed algorithm is always the best, which it’s not--the algorithm is only guaranteed to be better in certain accuracy as discussed in Section 4. Therefore, **the experiments illustrate the advantage suggested by our theory**, which is why we use the theoretical stepsizes. The same reasoning was applied to the federated learning experiments too. Optimal tuning of federated algorithms is a nontrivial problem on its own and includes tuning on both client and server level as studied by [Reddi et al., “Adaptive Federated Optimization”], so it is far beyond the scope of our work.

---

> > ### Author Response · Authors · 2021-08-19
> > **Complaint about Reviewer Gcuo**
> >
> > Dear AC, {cc: Reviewer Gcuo}
> >
> > As you will see, the reviewer praised our work in several ways. He/she only raises three pieces of criticism:
> >
> > 1) First, the author seems to think our work is a combination of Mishchenko et al (2020) RR method with standard prox analysis. We explain in our rebuttal that this is *not* the case. So, **this criticism is invalid.**
> >
> > 2) Second, the author suggests a theoretical comparison to FedAc. **We explained that our method is asymptotically faster.**
> >
> > 3) Lastly, the reviewer suggests we improve our experiments. **However, our work is theoretical, solving important open problems (i) can one design an RR method able to handle a prox term?, ii) can local methods beat GD on heterogeneous data?), and no experiments are needed for such work.** Still, we provided experiments which illustrate and confirm the consequences of our theory. This, we maintain, is wholly sufficient for a theory paper.
> >
> > We hope the reviewer will engage with us, but so far this did not happen. Please can you encourage him/her to reply to our rebuttal?
> >
> > We are convinced that score 5 is a gross underestimation of the contribution of our work.
> >
> > Thanks!!!
> >
> > Kind regards,
> >
> > Authors

---

> > ### Comment · Reviewer_Gcuo · 2021-08-19
> > **Response**
> >
> > While I appreciate that the authors have highlighted their results in the nonconvex setting, I still am not convinced that using theoretical stepsizes is the appropriate choice for conducting fair experimental comparisons. Thus, I stand by my score, as I believe it is a reasonable assessment of the work.

---

> > > ### Author Response · Authors · 2021-08-19
> > > **Re: "I still am not convinced that using theoretical stepsizes is the appropriate choice for conducting fair experimental comparisons"**
> > >
> > > Reviewer Gcuo and Dear AC,
> > >
> > > This reviewer raised 3 pieces of criticism in his/her original review: 1 - main criticism, and 2+3 - minor criticism (**labeled as minor by the reviewer!**)
> > >
> > > As we hope is
> > >
> > > - explicitly clear (re point 1: "...I appreciate that the authors have highlighted their results in the nonconvex setting") and
> > > - implicitly clear (re point 2: via no further comment provided)
> > >
> > > from the reply called "Response" to which we respond here, **we have addressed concerns 1 and 2 satisfactorily.**
> > >
> > > ---
> > >
> > > So, the only remaining point left is criticism 3 - which is **minor** in the words of the reviewer! Even if criticism 3 was valid (and we argue below it is not; and this should be obvious to any theoretician), we are fully convinced that it **should not be possible for any reviewer to keep his/her score unchanged** in such circumstances. Let us reiterate: **it should not be possible for any reviewer to suggest as low a score as 5 without any major criticism left!**
> > >
> > > If this review is taken at face value, this would highlight one of the ways in which the NeurIPS review system became infamous. Of course, we have faith that the reviewer will change his/her mind, or that the AC or SAC/PCs will step in to remedy this sorry situation.
> > >
> > > ---
> > >
> > > Re criticism 3:
> > > > I still am not convinced that using theoretical stepsizes is the appropriate choice for conducting fair experimental comparisons
> > >
> > > Our work is theoretical in nature, and it stands on its own without **any** experiments whatsoever.
> > > - It should not possible for a top venue such as NeurIPS to accept purely experimental work without any theory, and at the same time reject purely theoretical work without any experiments. Of course, theory and practice are both equally important.
> > > - Moreover, we **do provide experiments**. And our experiments serve the fully justified and deeply scientific purpose: **to verify that our theory has predictions that can be observed in practice.** That is, to illustrate that our key theoretical predictions can be verified through experimentation. But in order to do that, we of course **absolutely need to rely on the theoretical stepsizes predicted by our theory! This is the fair and correct thing to do.**
> > > - Now, of course, **practitioners must go beyond theory in many ways if they want to achieve SOTA practical performance.** One such trick involves the *tuning* of stepsizes (which, in its basic form, is simply just a practical heuristic). Our goal in this paper is *not* to produce fine-tuned software. We do not therefore need to be preoccupied with such questions. That is not what our paper is about. **We believe the author is applying a standard to our work that simply can't be applied here.** Was is happening is this: the reviewer ignores our theory altogether (how else can we explain a score of 5 with no theory issues outstanding?), and focuses on a very small aspect of practical importance, labeled as minor by the reviewer, which has a role in computation-oriented works, but is irrelevant in our work.
> > >
> > > ---
> > >
> > > **We hope the reviewer will dramatically increase the score. If this does not happen, we kindly request the AC to disregard this review altogether.**
> > >
> > > Thank you,
> > >
> > > Authors

---

### Official Review · Reviewer_X7ht · 2021-07-23

**Rating:** 6
**Confidence:** 4

**Summary:**

This paper proposed two shuffling based stochastic optimization methods, ProxRR and FedRR. ProxRR is the proximal version of SGD with shuffling, and they prove that ProxRR achieves better convergence rate than vanilla proximal SGD on strongly convex function, similar to its counterpart SGD with shuffling. Another advantage of this paper is that their convergence results do not need bounded gradient assumption, which was needed in the several prior shuffling SGD papers. In FedRR algorithm, each client uses SGD without replacement to optimize the local model, instead of vanilla SGD employed in local SGD. They also have analysis of FedRR for convex and nonconvex functions under both homogeneous and heterogeneous settings, and the convergence rates are faster than local SGD. The experiments also show the faster convergence of shuffling based algorithms.


**Limitations And Societal Impact:**

1. For ProxRR, I am curious about the comparison with the literature of shuffling SGD. Particularly, [1] gives a lower bound $O(n/T^2)$, so can we choose the learning rate carefully such that we can match with this lower bound?

2.  It is a decent algorithm and it does provably improve local SGD. My minor concern is that FedRR is a simple combination of local SGD and SGD without replacement. It would be great if we can apply the shuffling idea on client level as well, e.g., for model averaging stage, server does not randomly sample a subset of clients to do the average, instead it shuffles the clients and selects subsets without replacement. It makes the algorithm look more tailored for federated learning.

[1] Rajput, Shashank, Anant Gupta, and Dimitris Papailiopoulos. "Closing the convergence gap of SGD without replacement." International Conference on Machine Learning. PMLR, 2020.


**Main Review:**

As far as I know, this paper is the first to analyze the convergence of proximal SGD without replacement, and it is also the first to introduce shuffling ideas into federated optimization. The theoretical analysis is sound to me. The experiments also support their theory. The paper is well-written and both experimental and theoretical results are clearly presented. This paper introduces shuffled local updates in local SGD, which is a well-known idea to speedup SGD in the traditional optimization community. As the authors showed in the paper, this idea does improve the convergence rate of vanilla local SGD, and the experiments also support that.

**Time Spent Reviewing:**

3

---

> ### Author Response · Authors · 2021-08-10
> **Thank you for the positive feedback!**
>
> We thank the reviewer for the positive feedback. We are glad that the reviewer appreciated our theoretical and numerical results as well as the idea itself. Below we address the limitations:
> 1. **Yes, we can!** When we specialize to the setting without proximal operator, for which the cited lower bound is given, **we recover the optimal rate** $O(n/T^2)$ (ignoring logarithmic factors), which is presented as an equivalent $O(\sqrt{n}/\sqrt{\varepsilon})$ rate. Note that for the case with proximal operator, there is no lower bound in the literature.
> 2. It is indeed a simple combination of **two highly influential algorithms**. Both local SGD and RR are at the core of machine learning and federated learning, so we believe that it’s by no means a drawback that our proposed algorithms are based on them. In some specialized cases, our guarantees match the lower bounds, so that the theory is also tight at least in the sense of one special case. We agree that client sampling strategies is an interesting potential extension, which one can analyze by viewing each client update as an approximate gradient step, and which should be possible. We would like to explore this direction, but our paper seems to already have too many results, for instance, reviewer Ak2z reviewed our work for 8 hours but still did not mention that Algorithm 1 has nonconvex guarantees too; none of the reviewers seemed to comment on our importance sampling strategies either. We believe that one first needs to build a solid theory for the simple case, which we do here, and only then work on more sophisticated extensions.
>
> It surprises us that the reviewer gave us **a lot of positive feedback** and the **lowest** possible accept score (6). We believe that the review corresponds to a higher score in the range (7)-(9). If the reviewer finds our answers satisfactory, the assessment, perhaps, corresponds even to a score in the range (8)-(10). It is important to us that the reviewer takes this into account as the papers are often judged based on the average score of the reviews rather than their content.

---

> ### Author Response · Authors · 2021-08-25
> **Please let us know if our response was helpful**
>
> Dear reviewer,
> We thank you again for your feedback. It's been 2 weeks since our response to you, and we are wondering if you have or are going to read it. Since you had a question in your original review that we answered, we hope that you would at least let us know if the answer was satisfactory.

---

### Author Response · Authors · 2021-08-19
**Reviewers: Please can we ask you to respond to our rebuttals?**

Dear reviewers,

We would wish to kindly ask you to respond to our rebuttals (which we submitted 8 days ago). We believe we have addressed all your concerns, and we would want to hear your views. If you agree that we *did* address your concerns, this should affect your scores. Please let us know. If you believe we have *not* addressed some of your concerns, please let us know *why*, so that we can either admit there is an issue somewhere, or so that we have a chance to further clarify.

**Let us all use the OpenReview platform the way it is supposed to be used: to enable reviewer-author dialogue so as to maximize the chances of understanding on both sides. We all know the review system at large AI/ML conferences has serious issues. The move from CMT to OpenReview was supposed to enable a dialogue as this was believed to be useful in generating a more fair and professional review process.**

We hope you share our view of what OpenReview should be, and we hope you will engage with us.

Thank you!

Kind regards,

Authors

---

### Decision · Program_Chairs · 2021-09-27

**Decision:**

Reject

**Comment:**

After all reviews, responses, and discussions, there is general agreement that the paper is clear and well motivated on its own, but that its main contributions in relation to previous work appear limited as currently presented. The concerns about improvements on prior work took several forms, with the most common being that the extension above Mishchenko et al. (2020) seemed incremental. The discussion clarified a few finer details about this, but in the end did not address the basic concerns of multiple expert reviewers, even considering the nonconvex analysis.

There was discussion on a minor point between Reviewer Gcuo and the authors about tuning step-sizes in experiments. I understand that step-sizes from analysis are used on purpose in order to simulate the theory. Although tuning these would be a different experiment, it could be a related interesting one. Regardless, I did not read this as the main point of that review and would not consider it a serious issue affecting the overall recommendation. It is clear that this paper focuses on theory.